# HNRNPL Restrains *miR-155* Targeting of BUB1 to Stabilize Aberrant Karyotypes of Transformed Cells in Chronic Lymphocytic Leukemia

**DOI:** 10.3390/cancers11040575

**Published:** 2019-04-23

**Authors:** Sara Pagotto, Angelo Veronese, Alessandra Soranno, Veronica Balatti, Alice Ramassone, Paolo E. Guanciali-Franchi, Giandomenico Palka, Idanna Innocenti, Francesco Autore, Laura Z. Rassenti, Thomas J. Kipps, Renato Mariani-Costantini, Luca Laurenti, Carlo M. Croce, Rosa Visone

**Affiliations:** 1Ageing Research Center and Translational medicine-CeSI-MeT, 66100 Chieti, Italy; sara.pagotto@unich.it (S.P.); a.veronese@unich.it (A.V.); alice.ramassone@unich.it (A.R.); rmc@unich.it (R.M.-C.); 2Department of Medical, Oral and Biotechnological Sciences, “G. d’Annunzio” University Chieti-Pescara, 66100 Chieti, Italy; alessandra.soranno@gmail.com (A.S.); paolo.guanciali@unich.it (P.E.G.-F.); giandomenico.palka@unich.it (G.P.); 3Department of Medicine and Aging Sciences, “G. d’Annunzio” University Chieti-Pescara, 66100 Chieti, Italy; 4Department of Cancer Biology and Genetics, Comprehensive Cancer Center, The Ohio State University, Columbus, OH 43210, USA; Veronica.Balatti@osumc.edu (V.B.); Carlo.Croce@osumc.edu (C.M.C.); 5Institute of Hematology, Catholic University of the Sacred Heart, 00168 Rome, Italy; idanna.innocenti@yahoo.it (I.I.); francesco_autore@yahoo.it (F.A.); Luca.Laurenti@unicatt.it (L.L.); 6Department of Medicine, Moores Cancer Center, University of California at San Diego, La Jolla, CA 92093, USA; lrassenti@ucsd.edu (L.Z.R.); tkipps@ucsd.edu (T.J.K.); 7Chronic Lymphocytic Leukemia Research Consortium, San Diego, CA 92093, USA

**Keywords:** *miR-155*, BUB1, HNRNPL, CIN, CLL and microRNA

## Abstract

Aneuploidy and overexpression of *hsa-miR-155-5p* (*miR-155*) characterize most solid and hematological malignancies. We recently demonstrated that *miR-155* sustains aneuploidy at early stages of in vitro cellular transformation. During in vitro transformation of normal human fibroblast, upregulation of *miR-155* downregulates spindle checkpoint proteins as the mitotic checkpoint serine/threonine kinase budding uninhibited by benzimidazoles 1 (BUB1), the centromere protein F (CENPF) and the zw10 kinetochore protein (ZW10), compromising the chromosome alignment at the metaphase plate and leading to aneuploidy in daughter cells. Here we show that the heterogeneous nuclear ribonucleoprotein L (HNRNPL) binds to the polymorphic marker D2S1888 at the 3′UTR of *BUB1* gene, impairs the *miR-155* targeting, and restores BUB1 expression in chronic lymphocytic leukemia. This mechanism occurs at advanced passages of cell transformation and allows the expansion of more favorable clones. Our findings have revealed, at least in part, the molecular mechanisms behind the chromosomal stabilization of cell lines and the concept that, to survive, tumor cells cannot continuously change their genetic heritage but need to stabilize the most suitable karyotype.

## 1. Introduction

*Hsa-miR-155-5p* (*miR-155*) contributes to genomic instability through downregulation of proteins implicated in DNA mismatch repair, DNA damage repair, and maintenance of telomere integrity [1,2,3]. We recently shed light on its involvement in the spindle assembly checkpoint (SAC) and chromosome instability (CIN) during the early stages of the in vitro transformation of human normal dermal fibroblasts (HDF) by hTERT and SV40 large T antigen-enforced expression [4,5,6]. We demonstrated that *miR-155* targets BUB1, CENPF and ZW10 [6], proteins implicated in preventing anaphase onset in case of unattached kinetochores [7,8]. BUB1, the most studied of the three, is a serine/threonine kinase localized at the outer plate of the kinetochore whose main function is to ensure the correct chromosome alignment of the metaphase plate [8]. Indeed, BUB1 deregulation is strongly associated to aneuploidy in mammalian cells [9,10,11]. Our data reflect the critical implications of CIN in tumor initiation [12,13,14,15], highlighting *miR-155* as a new player during tumor onset.

On the other hand, how CIN impacts the forward phases of tumor evolution is debated. Indeed, it is not clear if there is a continuous increase of different chromosomal aberrations over time, with the consequent positive selection of the fittest clones (“gradual” model); or if all the chromosomal aberrations are generated in a short period of time followed by expansion of the most stable clones (“crisis and stasis” model) [16,17]. The occurrence of equilibrium between the two mentioned models could also be a plausible paradigm. However, how in the “gradual” model the fittest clones could maintain a CIN behavior and continuously expand remains obscure. In this view, the “crisis and stasis” paradigm seems more appropriate considering the stasis phase is a more stable genomic condition.

To better investigate this assumption, we evaluated CIN associated with the *miR-155*/BUB1 axis at the late passages of our HDF immortalized models and in a cohort of patients with chronic lymphocytic leukemia. We also investigated the mechanisms that could interfere with *miR-155*/BUB1 targeting, such as the involvement of HNRNPL, an RNA-binding protein that hampers miRNAs’ action by binding the target site on the miRNA [18,19].

## 2. Results

### 2.1. Role of miR-155 in Advanced Passages of Immortalized HDF Cells

To explore to what extent *miR-155* affects aneuploidy in more advanced stages of transformation, we analyzed chromosomes at different passages during the immortalization process of normal HDF cells (HDF_LT/hTERT_ cells at passages P16, P20 and P24) and found a similar number of karyotype abnormalities at P16 and P20, but a higher number at P24 (Figure 1A). We next evaluated the morphology of the metaphase plates of the same cells by immunofluorescence, considering abnormal those metaphases that showed chromosomes with distinct spindle-positioning defects and incomplete congression [6]. We found an initially high rate of abnormal metaphase (P16) followed by a reduction at P20 and P24 (Figure 1B). To better understand this apparent discrepancy, i.e., a higher number of aberrant karyotypes with normal metaphase plates, we grouped the identified karyotypes by separating cells with unique karyotypes from cells sharing a common karyotype or related karyotypes (considered sub-clonal). The number of such clones and related sub-clones evolves from P16 to P24, whereas single cells with aberrant karyotypes decrease (Figure 1C; Appendix A). In summary, the most adaptive clones expand during these passages, keeping normal metaphase plates. Considering these results, we measured BUB1 protein expression during the multistep immortalization of HDF_LT/hTERT_ cells (from P14 to P24). In two of the three experiments, Western blotting showed an increase in BUB1 protein, despite the steadily high *miR-155* expression (Figure 1D and Appendix A), suggesting a possible mechanism of escape from *miR-155* targeting.

### 2.2. The RNA-Binding Protein HNRNPL Correlates Positively with BUB1 in CLL Cells

To evaluate the exclusivity of this *miR-155*/BUB1 targeting escape we tested in the same Western blot two other known *miR-155* target proteins: mutL homolog 1 (MLH1) and mutS homolog 6 (MSH6) [1]. We found a recurrent modulation of these proteins during the passages (Appendix A), supporting the hypothesis that *miR-155* keeps its targeting function on those genes but loses that on *BUB1*. To test this hypothesis, we first analyzed the 3′UTR sequence of *BUB1* and we found a CA repeat sequence ((CA)_n_), also known as the polymorphic marker D2S1888 (rs113391170 at the https://www.ncbi.nlm.nih.gov/snp). We focused on this marker because: (1) *miR-155* is involved in microsatellite instability (MSI) in colorectal cancer by targeting hMLH1, MSH2 and MSH6 and therefore potentially affecting the CA repeat sequence [1], and (2) the CA repeat may enable *BUB1* mRNA to elude miRNA targeting by binding the RNA-binding protein HNRNPL depending on the length of the marker [18,19]. We measured the length of the dinucleotide, at the genomic level, during the transformation of HDF_LT/hTERT_ cells by fragment length analysis (FLA), and observed an induced instability from P19 to the last passage analyzed during HDF immortalization (Appendix A). Over advanced passages of HDF_LT/hTERT,_ we observed that the instability of the polymorphic CA repeat correlated with increased BUB1 protein levels, suggesting that this mechanism affects later passages of HDF immortalization. However, in the other two independent experiments, despite the increased BUB1 expression in one of them, we did not register any instability of the D2S1888 marker over the cell line passages. To better understand the role of this genetic locus in the regulation of BUB1 expression, we next focused on human tumors, specifically on chronic lymphocytic leukemia (CLL), for the following reasons: (1) the 2q13 chromosome region where *BUB1* maps is associated with CLL risk [20]; (2) we found two missense germline mutations in the *BUB1* coding sequence in two patients of a cohort of 30 CLL patients carrying 11q deletions (Figure 2); (3) BUB1 targeting by *miR-155* was also observed in a CLL cell line [6]; (4) HNRNPL binds the 3′UTR of *BUB1* at the D2S1888 polymorphic marker in lymphoid cells (Appendix A) [21]. We investigated by FLA the D2S1888 marker in DNA from PBMC of 172 CLL patients, 20 of whom had two longitudinal points (for a total of 212 samples), and 86 healthy donors, and found no significant differences in allele or homozygous frequencies between these cohorts (Appendix A). Then, given that B-CLL cells are the principal altered component in CLL, we compared the D2S1888 genetic profiles of sorted CD3^+^ CD16^+^ cells and leukemic cells from total PBMCs of the same patient and at two longitudinal points in two of the five CLL patients with progressive disease. As shown in Appendix A, we did not observe any differences in D2S1888 between healthy and leukemic cell fractions nor during disease progression. As previously mentioned, miRNA targeting could be regulated by the binding of HNRNPL to CA repeats [18], and the affinity of HNRNPL to its RNA target is directly proportional to the length of the (CA)_n_ [22]. However, considering the stability of D2S1888, we evaluated HNRNPL and BUB1 protein expression in longitudinal samples of PBMCs or pure B-CLL cells from 13 CLL patients (Appendix A). When the content of B cells was low or different between the sequential PBMC samples (CD5/CD19 < 70% of PBMC), B cells were purified. As shown in Figure 3, nine patients (70%) had an increase in BUB1 protein levels over time. When we evaluated the relative *miR-155* expression, except for patients LLC05, we did not find correlations with BUB1 protein levels between the longitudinal time points. These data are in line with the above-presented results on HDF_LT/hTERT_ cells (Figure 1D). However, we found a clear, positive correlation between BUB1 and HNRNPL protein expression in 12 out of 13 patients (92%; ρ = 0.6630; *p* = 0.0116) (Figure 3 and Appendix A).

### 2.3. HNRNPL Interaction on 3′UTR of BUB1 Inhibits miR-155 Targeting

To determine if HNRNPL interacts with *BUB1* 3′UTR in B-CLL cells as occur in T cells (Appendix A), we performed RNA immunoprecipitation (RIP) analysis in HG-3 cells, a CLL cell line in which we transfected the vectors carrying the *BUB1* 3′UTR with either 18CA or 19CA (the most representative alleles) downstream of the luciferase reporter gene. This analysis showed that specific binding of HNRNPL to the untranslated region of *BUB1* RNA was stronger for 19CA than for 18CA (Figure 4A), as previously reported in a different cellular context [22]. We also observed, in two independent experiments, that HNRNPL silencing was correlated with a decrease in BUB1 protein expression in HG-3 cells, while *BUB1* mRNA levels did not change (Figure 4B and Appendix A). Thus, we propose that HNRNPL plays a significant role in the post-transcriptional regulation of BUB1. To define the interplay among *miR-155*, BUB1 and HNRNPL, we investigated the influence of *miR-155* on the *BUB1* 3′UTR after silencing HNRNPL in HG-3 cells. By inhibiting *miR-155*, the luciferase activity of the *BUB1* 3′UTR-reporter gene construct increased (~15%, *p* = 0.0034) only in HNRNPL-depleted cells (Figure 4C). To further investigate the *BUB1* 3′UTR expression, we analyzed by RT-PCR the 522 bp amplicon of the 3′UTR of *BUB1* on HG-3 cells depleted of HNRNPL. We noticed an additional shorter PCR product of about 160 bp in both HNRNPL-depleted and control cells (Figure 5A). This PCR product was also observed in our HDF_LT/hTERT_ cells during the immortalization, therefore cloned, sequenced and aligned to the genomic sequence of *BUB1* gene (Figure 5B,C). We identified a new spliced variant of the 3′UTR of *BUB1*, excluding the D2S1888 genetic locus but retaining the miR-155 binding site. To further establish the presence of this spliced variant, we designed specific PCR primers (u58_F-4090_R; Figure 5C; Appendix A) able to identify only this form to test its expression in eight PBMC sequential samples from four different CLL patients. All the samples present the specific PCR product of the spliced 3′UTR (Figure 5D). We analyzed by luciferase assay the susceptibility of the construct containing the spliced form (BUB1 3′UTR SPL) to the *miR-155*. The luciferase activity increased about 100% (*p* < 0.0001) after ectopic expression of miR-155 antisense inhibitor (miR-155AS) on MEC-1 cells and decreased over 100% (*p* < 0.0001) after *miR-155* overexpression on HDF_LT/hTERT_ cells (Figure 6A). This demonstrated the interaction between the new variant of the 3′UTR of *BUB1* and the *miR-155*, suggesting that the post-transcriptional regulation of BUB1 is influenced by the local mRNA structure. To assess this hypothesis, we evaluated changes in BUB1 protein levels by Western blotting in two different B-CLL cell lines after miRNA or HNRPL loss of function using a “sponge” technique. Briefly, HG-3 and MEC-1 cell lines were transfected with increasing doses of luciferase expression vectors carrying either the spliced form or the full *BUB1* 3′UTR. As shown in Figure 6B, the BUB1 3′UTR SPL vector, acting as a *miR155*-sponge, increases the endogenous BUB1 protein expression, derived by the spliced form and/or by *BUB1* mRNA not bound to HNRNPL, in a vector dose-dependent manner in both cell lines. On the other hand, the luciferase vector containing the full *BUB1* 3′UTR acting as an HNRNPL-sponge showed an essential reduction of the endogenous BUB1 expression with no evident dose-dependent behavior. In this case depletion of HNRNPL probably permits the post-transcriptional regulation of BUB1 by the *miR-155* in both *BUB1* mRNA forms, although we cannot exclude a different *miR-155* targeting of the two 3′UTR identified (Figure 7).

## 3. Discussion

Chromosomal abnormalities are present in 60–80% of the human tumors [23,24], making CIN one of the most studied hallmark of cancer. CIN is positively correlated with tumor stage, relapse, metastasis, treatment and resistance [25,26], and is predominately caused by defects in the mitotic chromosome segregation [27]. However, despite the relevance of CIN in cancer, its role in the evolution of tumors seems contradictory since an excessive and extensive CIN is incompatible with life, even for neoplastic cells [28]. To explain this phenomenon seems indispensable to optimize tumor fitness by attuning CIN, with an intricate interplay between cellular selection and molecular activation [29,30].

We recently described how *mir-155* targets BUB1, CENPF and ZW10 during the early stages of cellular transformation and causes cellular aneuploidy [6]. However, at more advanced stages of in vitro transformation, we witness an increase of normal metaphases despite the expansion of cells with abnormal karyotypes and the consistently high expression of *miR-155*. To explain this apparent discrepancy, we supposed a possible *miR-155* two-step mechanism to sustain cellular transformation: initially its upregulation induces chromosome instability targeting genes of the mismatch repair, such as *hMLH1*, *MSH2* and *MSH6* [1], and genes involved in the SAC of the cell cycle, such as *BUB1*, *CENPF* and *ZW10* [6]; then, once we induced the fittest genetic abnormalities, a second step of genomic stabilization can be supported by inducing MSI at the 3′UTR of *BUB1* gene to impair *miR-155* targeting and restore BUB1 expression. However, this hypothesis was not verified in our CLL model, which, to restore BUB1 expression, uses another mechanism or rather increases the expression of the RNA binding protein HNRNPL, which binds the *BUB1* 3′UTR and impairs *miR-155* targeting. Therefore, in B-CLL cells where *miR-155* is upregulated [31], BUB1 and HNRNPL are positively correlated, and mostly they both increase over time (Figure 3). Our data show the importance of BUB1 to maintain the stability of aneuploidy cancer cells, and are in line with papers that show (i) the key role of BUB1 in maintaining human cell viability in the near haploid cell line HAP-1, which is derived from the chronic myeloid leukemia KBM-7 cell line [32]; (ii) the robust spindle check point of aneuploid colon cancer cell lines [33]; and (iii) the capacity to sensitize tumor cells to taxanes, ATR and PARP inhibitors by hindering BUB1 activity in triple-negative breast cancer in vitro and in vivo models [34].

## 4. Materials and Methods

### 4.1. Primary Cells

Blood samples from healthy donors (HD) were obtained at the “Centro di Riferimento Regionale di Genetica Medica” (Chieti, Italy), while blood samples from Chronic Lymphocytic Leukemia (CLL) patients were obtained from “Policlinico Universitario Agostino Gemelli” (Rome, Italy) and biobanks of the CLL Research Consortium (UCSD, San Diego, CA, USA). HD and CLL patients gave written informed consent for the use of their samples for research purposes, in accordance with the Declaration of Helsinki. The study protocols were approved by the institutional review board of the “G. d’Annunzio” University of Chieti-Pescara (18_verb.12.07.2018), the “Policlinico Universitario Agostino Gemelli” of Rome (P/948/CE/2011; P/392/CE/2012; 0015023/16) and UCSD (171884CX), respectively.

Peripheral blood mononuclear cells (PBMC) were isolated by density gradient centrifugation with Ficoll-Paque Plus (GE Healthcare, Chicago, IL, USA). B-CLL cells purification was performed using B Cell Isolation Kit II, human (Miltenyi Biotec, Bergisch Gladbach, Germany), according to the manufacturer’s instructions.

### 4.2. DNA/RNA Extraction and Reverse Transcription Quantitative PCR (RT-qPCR)

DNA was extracted from cell lines, blood samples, PBMC and B-CLL cells using the PureLink^®^ Genomic DNA Kit (Thermo-Fisher Scientific, Waltham, MA, USA), QIAamp DNA Blood Kit (Qiagen, Hilden, Denmark), or by Phenol-chloroform extraction. DNA quality was assessed by gel electrophoresis on 0.8% agarose gel electrophoresis in 0.5× Tris borate EDTA (TBE) buffer in the presence of ethidium bromide. Total RNA was isolated from cells using QIAzol Lysis Reagent (Qiagen, Hilden, Denmark) according to the manufacturer’s instructions. RNA quantification was performed by NanoDrop 2000 (Thermo-Fisher Scientific, Waltham, MA, USA). *MiR-155* was reverted from 25 ng of total RNA using specific reverse primers (stem loop RT primers designed with modifications to include the UPL #21 (Roche, Basel, Switzerland, complementary sequences; Appendix A) and the TaqMan Micro-RNA Reverse Transcription Kit (Thermo-Fisher Scientific, Waltham, MA, USA). Reactions were incubated 30 min at 16 °C, followed by pulsed RT of 60 cycles at 30 °C for 30 s, 42 °C for 30 s, and 50 °C for 1 s [35]. Reverse transcription quantitative PCR was performed on the CFX96 Touch™ Real-Time PCR Detection System (Bio-Rad, Hercules, CA, USA), using the Universal Mastermix (Roche, Basel, Switzerland). Target amount, normalized to the endogenous reference RNU44 was determined using the 2^−Δct^ method (User Bulletin #2, Applied Biosystems, Thermo-Fisher Scientific, Waltham, MA, USA).

### 4.3. Cell Culture

MEC-1 and HG-3 cell lines were acquired from Leibniz Institut DSMZ—the German Collection of Microorganisms and Cell Cultures, Inhoffenstraße 7B, 38124 Braunschweig, Germany- (December 2015 and February 2016, respectively). Cells were cultured in RPMI medium supplemented with 10% fetal bovine serum, 1% Pen/Strep and 1% L-glutamine (Sigma-Aldrich, Saint Louis, MO, USA) at 37 °C in a 5% CO_2_ incubator. Normal human adult dermal fibroblasts (HDFa, Thermo-Fisher Scientific, Waltham, MA, USA) were immortalized using retroviruses (RV LTSV40 and RV hTERT) in combination with antisense of *miR-155* or control lentiviruses (LV AS *miR-155*, LV CTRL), as described in Pagotto et al. [6]. Briefly, normal HDF cells were infected at passage #4 (P4), with the retrovirus transducing the SV40 Large T antigen for 3 h, then they were infected overnight with LV pmiRZip AS miR-155 and LV pmiRZip CTRL (System Biosciences, Palo Alto, CA, USA). Then, the cells were selected for seven days with puromycin (0.5 μg/mL) (Santa Cruz Biotechnology, Dallas, TX, USA). After selection the cells were infected at passage #6 (P6) with the retrovirus transducing the hTERT gene for 3 h. Then hygromycin selection was performed for an additional seven days (50 μg/mL) (InvivoGen, San Diego, CA, USA). Since LV AS CTRL contains the copGFP reporter gene, GFP-positive cells were sorted using the FACSAriaIII, 100 μm nozzle (BD Biosciences, Becton, Dickinson and Company, Franklin Lakes, NJ, USA). Immortalized HDF cells infected with LV AS *miR-155* were not considered in this paper whereas the LV CTRL immortalized HDF cells were named: HDF_LT/hTERT._ HDF_LT/hTERT_ cells were cultured in Medium 106 (M106) supplemented with Low Growth Serum Supplement (LSGS) (Thermo-Fisher Scientific, Waltham, MA, USA).

### 4.4. Plasmids

The spliced 3′UTR region of *BUB1* gene was amplified by PCR from HDF genomic DNA and cloned into a pGL3-Control firefly luciferase reporter vector using the XbaI restriction enzyme immediately downstream of the luciferase gene (pGL3_3′UTR BUB1 SPL, Promega, Madison, WI, USA). The 3′UTR regions of *BUB1* gene with different CA repeats were cloned downstream the *Renilla* luciferase gene in the psiCHECK_2 vector (psiCHECK-2_3′UTR BUB1 (CA)_n_, Promega, Madison, WI, USA), as described in Pagotto et al. [6].

### 4.5. Cells Transfection and Luciferase Target Assays

Transfections of MEC-1 and HG-3 cells were performed using Amaxa™ Nucleofector™ II (Lonza, Basel, Switzerland), program U-016; U-023 for HDF_LT/hTERT_ cells. Briefly, 1–5 × 10^6^ cells were transfected with plasmid DNA (2 µg/L Million of cells); and/or with the antisense inhibitor oligonucleotide, AS *miR-155*: ID:AM12601 (200 nM final concentration in the culture media); or *miR-155* precursor (100 nM final concentration in the culture media), ID:PM12601 and respective negative controls AS NC and NC (Thermo-Fisher Scientific, Waltham, MA, USA)). The HNRNPL knockdown gene expression was performed on HG-3 cells using small interfering RNAs (hNRNPL siRNA (h), or Control-siRNA-A, Santa Cruz Biotechnology, Dallas, TX, USA) at a final concentration of 200 nM in 2 mL of culture media per well/sample. Firefly and *Renilla* luciferase activities were measured using the Dual-Glo Luciferase Assay System (Promega, Madison, WI, USA) according to the manufacturer’s instructions. To control the transfection efficiency, *Renilla* luciferase activity was normalized to firefly luciferase activity.

### 4.6. Mutational Analysis

DNA sequencing was performed using the Big Dye Terminator v3.1 (Thermo-Fisher Scientific, Waltham, MA, USA) in an automated DNA capillary ABI 3130 XL sequencing machine (Applied Biosystems, Thermo-Fisher Scientific, Waltham, MA, USA). Primers were designed to amplify all the exons of *BUB1* and the regions surrounding them (Appendix A), and for the identification of mutations the sequenced data were compared against published *BUB1* gene sequences in GenBank.

### 4.7. Western Blotting

Cells samples were washed two times with cold PBS, and lysed in cold NIH buffer (1% Igepal, 1 mM EDTA, 50 mM TrisHCl, 150 mM NaCl) supplemented with proteases and phosphatase inhibitors (Protease Cocktail, Phosphatase Inhibitor Cocktail, Sigma-Aldrich, Saint Louis, MI, USA). Cell lysates were cleared by centrifugation (15 min at 14,000× *g* at 4 °C), then loaded on a Criterion TGX 4–20% precast polyacrylamide gel (Bio-Rad, Hercules, CA, USA) and transferred onto a PVDF membrane (Bio-Rad, Hercules, CA, USA). Primary antibodies used were: anti-BUB1 (B3), anti-γ-tubulin (C-20), anti-hnRNPL (4D11) (Santa Cruz Biotechnology, Dallas, TX, USA) and β-actin (Sigma-Aldrich, Saint Louis, MI, USA). Expression was detected using ECL chemiluminescent substrate (Thermo-Fisher Scientific, Waltham, MA, USA) after treating with specific HRP-conjugated secondary antibody (Cell Signaling Technology, Danvers, MA, USA). To quantify Western blot signals, high resolution of autoradiography digital images were acquired and band signals were quantified by ImageJ (Fiji software, 1.51 version, https://fiji.sc/).

### 4.8. Karyotype Analyses

To prepare metaphase spreads, 1–2 × 10^6^ HDF_LT/hTERT_ cells of different passages (P16/P20/P24) were plated onto 35-mm dishes with glass slides (Amniodishes, Euroclone, Pero, Milano, Italy), respectively. After 24 h, was carried out the colcemid treatment (Irvine Scientific, Santa Ana, CA, USA): 0.5 µg/mL for 4 h at 37 °C in an atmosphere of 5% CO_2_ and 20% humidity. Cells were then fixed in Carnoy’s solution (75% methanol, 25% acetic acid) and stained with Giemsa solution (5% for 10 min). Digital images were analyzed by Genikon software (3.8 version, Nikon, Shinagawa, Tokyo, Japan). The graphic representation of changes of sub-clonal populations over time was performed using the fishplot package for R [36].

### 4.9. Immunofluorescence

To visualize aberrant metaphase plates, HDF_LT/hTERT_ cells at different passages (P16/P20/P24) were seeded at mid-confluence in eight-well BD Falcon™ CultureSlides (7 × 10^3^ cells for well) (BD Bioscience, Becton, Dickinson and Company, Franklin Lakes, NJ, USA). Cells were arrested at pro-metaphase with 75 ng/mL of nocodazole (Sigma-Aldrich, Saint Louis, MI, USA) for 17 h plus 30 min of release, and prepared for immunofluorescence analysis as previously described in Pagotto et al. 2018 [6]. More than 150 images of metaphases were randomly acquired. Primary antibodies: anti alpha-tubulin (ab7291, Abcam, Cambridge Science Park, UK) and human anti-CREST (Antibodies Inc., Davis, CA, USA). Secondary antibodies: cross-adsorbed species-specific fluorescent secondary antibodies (Bethyl Laboratories, Montgomery, TX, USA). DNA was detected using DRAQ5 (Cell Signaling Technology, Danvers, MA, USA). All the images were acquired using a Carl Zeiss LSM5 Pascal confocal laser scanning microscope (Carl Zeiss 63× Plan Neofluar oil-immersion objective, Oberkochen, Germany).

### 4.10. RNA-Binding Protein Immunoprecipitation (RIP)

RIP was performed with Magna RIP kit (Millipore, Burlington, MA, USA). The assay was performed essentially according to the manufacturer’s instructions using 17 million HG-3 cells as the starting material. Magnetic beads (50 μL) were incubated for 30 min at room temperature with 5 μg of anti-HNRNPL antibody (4D11) or anti-WT1 antibody (F6) as negative control (Santa Cruz Biotechnology, Dallas, TX, USA), and then overnight with 100 μL of cell lysate at 4 °C. RNA was extracted using the miRNeasy Micro Kit and RNase-Free DNase Set (Qiagen, Hilden, Denmark) to perform a DNase digestion during RNA purification. The precipitated RNA was reverse-transcribed with and without reverse transcriptase to exclude plasmid DNA contamination. RT-qPCR was normalized to the non-IP control (input).

### 4.11. DNA Fragment Length Analysis (FLA)

FLA of DNA from 172 CLL patients, 20 of whom had two longitudinal points (for a total of 212 samples), 86 healthy donors, and HDF_LT/hTERT_ cells at different passages was performed using fluorescent D2S1888F_6FAM labeled forward primer and unlabeled reverse primer BUB1_3759 (Appendix A) on an ABI 3130 XL sequencing machine with ABI Standard Dye Set DS-33 and the GeneScan™ 500 LIZ^®^ (Thermo-Fisher Scientific, Waltham, MA, USA).

### 4.12. Statistical Analyses

Data regarding RT-qPCR, RIP and luciferase assays were evaluated using two-tailed Student’s *t*-tests, based on the significance obtained by the Shapiro‒Wilk Normality Test. Analysis of correlation between BUB1 and HNRNPL protein was performed by non-parametric Spearman’s correlation, two-tailed. All RT-qPCR assays were performed in triplicate. A *p*-value > 0.05 was considered not significant (ns); the significant *p*-values are reported in the figures. When indicated, standard deviation (SD) is represented by a scale bar on graphs.

## 5. Conclusions

We unraveled an important role of BUB1 in CLL, underlined by two germline mutations in the *BUB1* coding sequence in a series of 30 CLL cases, and demonstrated that BUB1 protein expression correlates with HNRNPL expression.

Our data support the hypothesis of a “tuning CIN to optimize tumor fitness” [29] by elucidating a novel molecular mechanism that highlight the role of the RNA binding protein HNRNPL as the tiebreaker for the *miR-155* targeting of BUB1. It is interesting to note that *miR-155* is an essential microRNA during inflammation [37,38], as was also evidenced for HNRNPL [39,40] and cellular aneuploidy [26,41], suggesting our finding as a novel dynamic molecular link between inflammation and cancer [42].

## Figures and Tables

**Figure 1 cancers-11-00575-f001:**
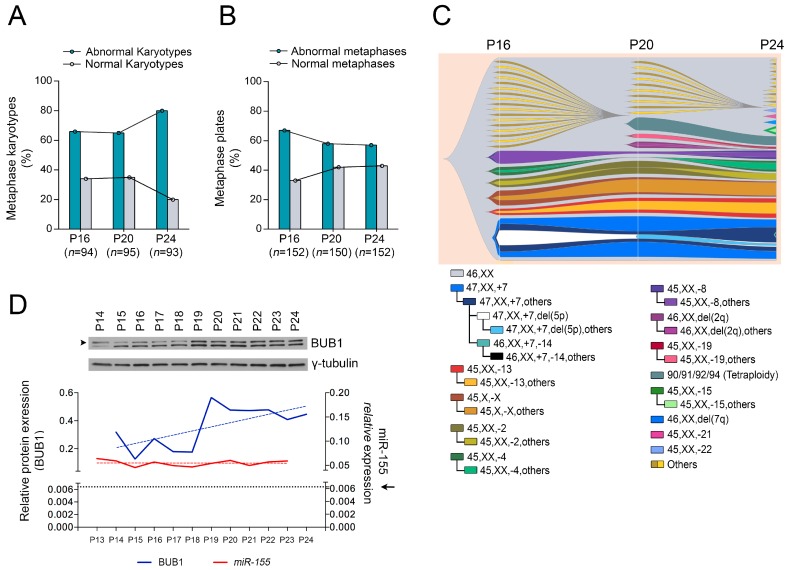
Expansion of most adaptive clones with normal metaphase plates is associated with increase of BUB1 expression in the advance passages of HDF_LT/hTERT_ cells. (**A**) Karyotype analysis and (**B**) metaphase analysis of HDF_LT/hTERT_ cells at P16, P20 and P24 passages. For karyotype analyses, HDF_LT/hTERT_ cells were arrested in mitosis by colcemid treatment (0.5 µg/mL for 4 h). A minimum of 93 metaphases was considered in each experiment. For immunofluorescence, the cells were arrested in mitosis by nocodazole (75 ng/mL) for 17 h, plus 30 min of release; >150 metaphases were considered for each experiment. We considered abnormal those metaphases that showed errors in chromosomes congression with distinct defects of the kinetochore alignment respect to the metaphase plate. The percentages calculated are the result of eight technical replicates. (**C**) Visualization of the karyotype analysis of HDF_LT/hTERT_ cell at P16, P20 and P24 passages with the fishplot package R. Each color represents a different karyotype identified at least twice in the metaphases analyzed (i.e., “46, XX”, normal karyotype; “47, XX, +7”, duplication of chromosome 7; “47, XX, +7, others”, duplication of chromosome 7 with others chromosome aberrations, etc.). Karyotype identified once were called “other karyotypes” and were represented in grey and light yellow. (**D**) Upper, Western blot analysis of BUB1 protein (120 kDa, blue line, P14-P24) and *miR-155* expression (red line, P13-P23) during the immortalization passages of HDF_LT/hTERT_ cells. Linear regression of *miR-155* (red dotted line) and of BUB1 protein expressions (blue dotted lines) are represented. The dotted line marked by the black arrow represents the *miR-155* expression level in normal HDF at P17. BUB1 protein levels were quantified by Western blotting using γ-tubulin as loading control.

**Figure 2 cancers-11-00575-f002:**
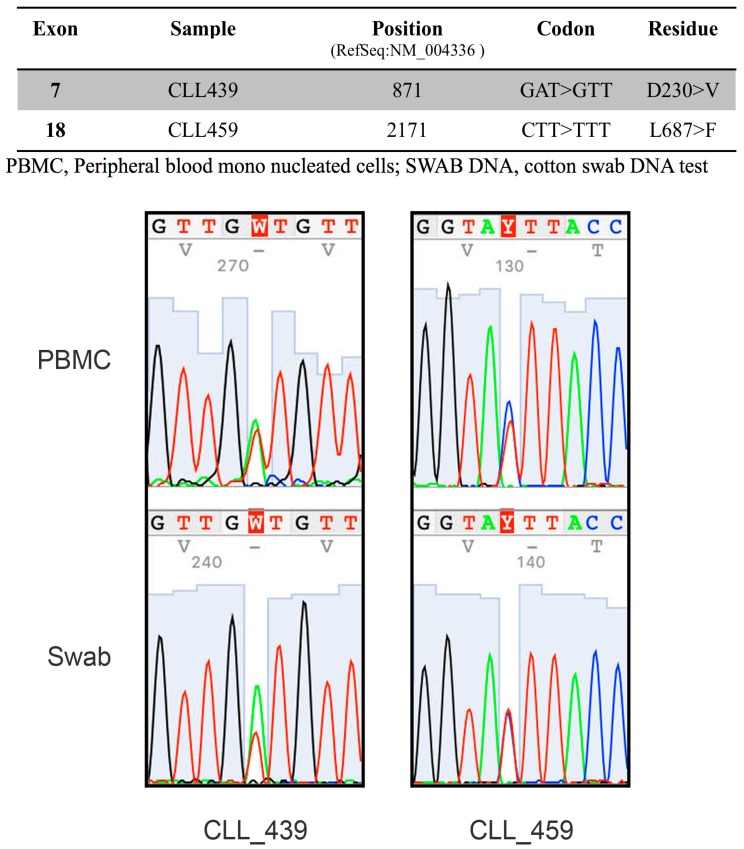
*BUB1* mutational analysis in CLL patients. Sequence analysis of *BUB1* from peripheral blood mononuclear cells (PBMC) and cotton buccal swab specimens (SWAB) genomic DNAs from two CLL patients with *BUB1* missense point mutations. Alignment was performed using the reference sequence NM_004336.

**Figure 3 cancers-11-00575-f003:**
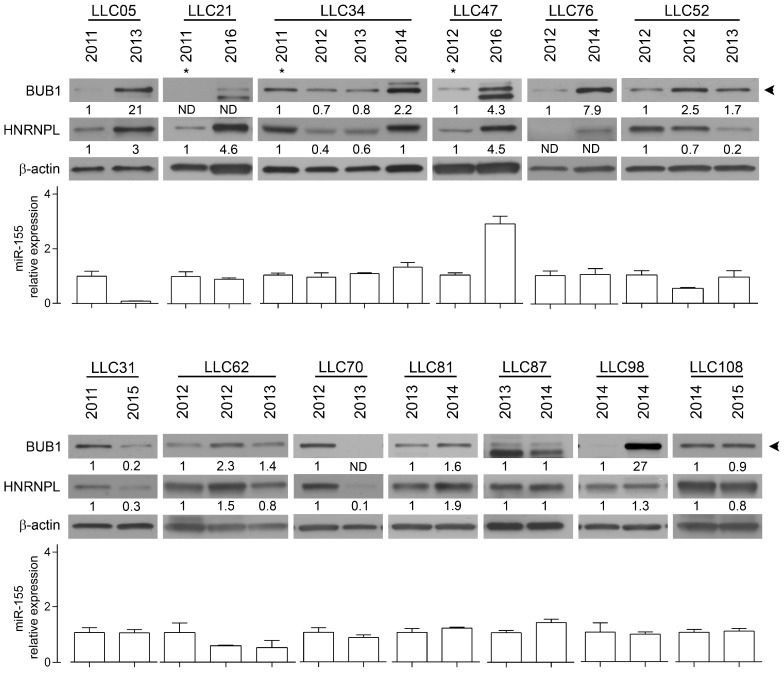
Positive correlation between BUB1 and HNRNPL protein expression in CLL patients. Western blots of BUB1 and HNRNPL proteins and RT-qPCR of *miR-155* in longitudinal samples of purified B-CLL cells (LL05, LLC21, LLC34, LLC47, LLC76) and PBMC with > 70% of CD5/CD19 positive cells (LLC31, LLC52, LLC62, LLC70, LLC81, LLC87, LLC98, LLC108) from CLL patients. Densitometric values normalized to β-actin expression are reported. Asterisks indicate time points with monoclonal B-cell lymphocytosis (MBL). *MiR-155* relative expression was normalized to the endogenous reference RNU44 (2^−Δct^ method). Data are means ± SD of three technical replicates.

**Figure 4 cancers-11-00575-f004:**
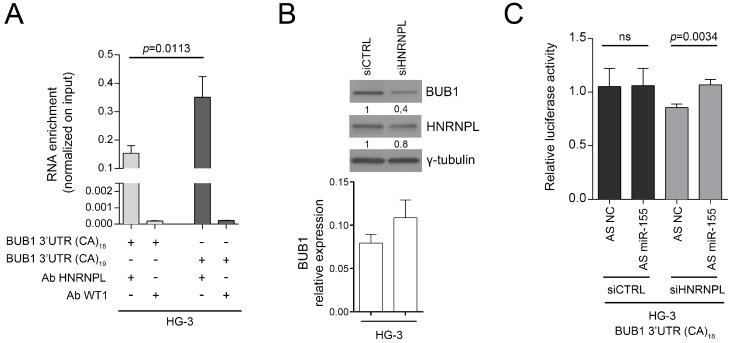
HNRNPL binding to CA repeats influences BUB1 targeting by *miR-155*. (**A**) *BUB1* 3′UTR mRNA quantification in RNA immunoprecipitated with either the anti-HNRNPL or the anti-WT1 antibody (as negative control) from HG-3 cells transfected as indicated. Data are means ± SD of three technical replicates normalized to the RNA input. *p*-value was calculated using the unpaired Student *t*-test, two-tailed. (**B**) Upper, protein expression levels of HNRNPL and BUB1 (120 kDa) in HG-3 cells transfected with either siRNA control (siCTRL) or siRNA of HNRNPL (siHNRNPL) for 72 h; bottom, *BUB1* mRNA relative expression was normalized to the endogenous reference β-actin (2^−Δct^ method). Data are means ± SD of three technical replicates (**C**) Luciferase assay in HG-3 cells transfected with siRNA control or siRNA of HNRNPL for 48 h, and subsequently re-transfected with psiCHEK_2 3′UTR BUB1 (18CA) vector in combination with anti-miR Negative Control (NC) or anti-miR-155 (AS miR-155) for additional 24 h. Data are means ± SD of three technical replicates of three independent transfections. The *p*-value was calculated using the unpaired Student *t*-test, two-tailed.

**Figure 5 cancers-11-00575-f005:**
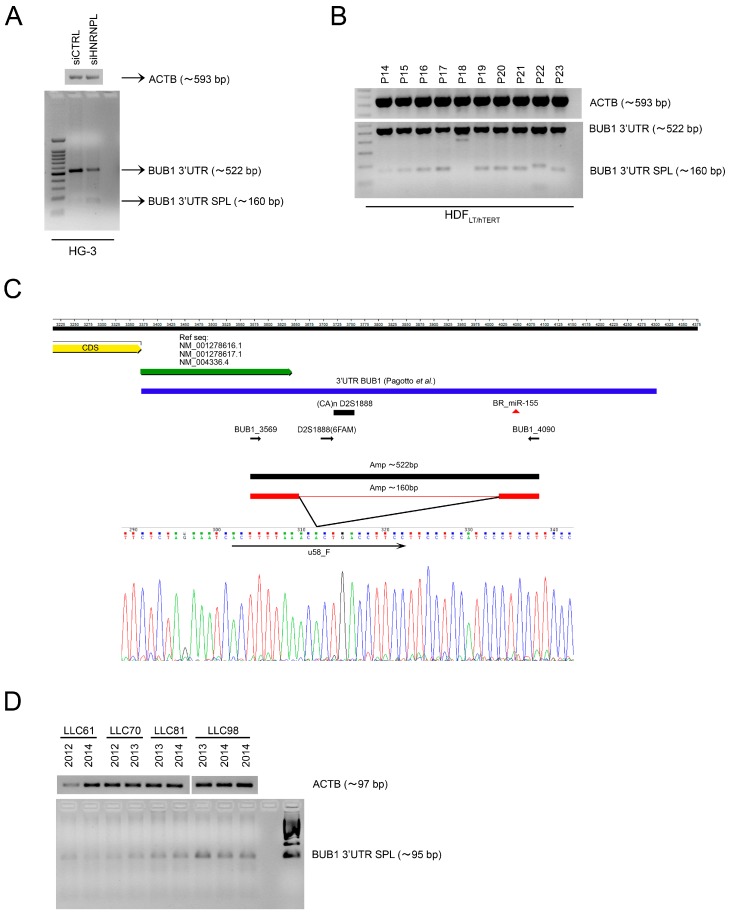
*BUB1* 3′UTR characterization. (**A**) Detection of the ~522 and ~160 bp *BUB1* 3′UTR amplicons by RT-PCR followed by gel electrophoresis in HG-3 cells after depletion of HNRNPL and (**B**) during the immortalization of HDF_LT/hTERT_ (1.5% agarose gel in 0.5× TBE buffer and ethidium bromide staining). (**C**) Schematic representation of *BUB1* 3′UTR shows, from top to bottom: 3′ end of the coding sequence (yellow arrow); NCBI/Ensembl reference 3′UTR sequence (green arrow); *BUB1* 3′UTR sequence identified in Pagotto et al. (blue box) [6]; D2S18888 polymorphic marker (black box); predicted miR-155 binding site (red triangle); and PCR primers (black arrows). On the bottom, sequence chromatograms of *BUB1* 3′UTR spliced form. (**D**) Detection of the ~95 bp amplicon of *BUB1* 3′UTR spliced form by RT-PCR followed by gel electrophoresis in eight PBMC sequential samples from four different CLL patients.

**Figure 6 cancers-11-00575-f006:**
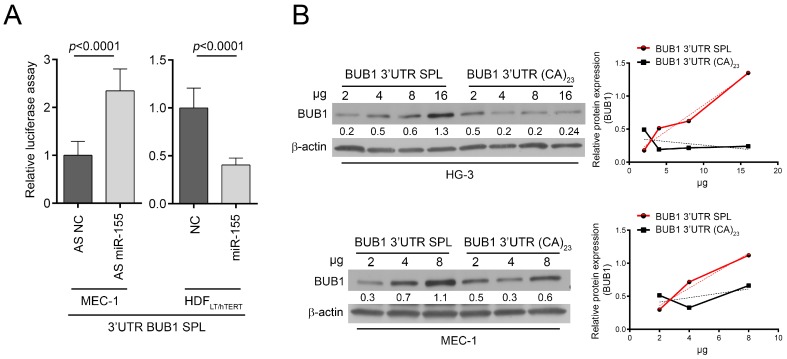
*miR-155* targeting of *BUB1* 3 ′UTR spliced form. (**A**) Relative luciferase activity in HG-3 and HDF_LT/hTERT_ cells after transfection with *miR-155* precursor (miR-155), antisense inhibitor oligonucleotide (AS miR-155), or respective negative controls (NC, negative control; and AS NC, anti-sense negative control) as indicated. Values are means ± SD of three technical replicates of two independent experiments. (**B**) Western blots of BUB1 in HG-3 and MEC-1 cell lines transfected with increasing doses of luciferase vectors carrying the spliced form (SPL) or the full *BUB1* 3′UTR (23CA). Densitometric values, normalized to β-actin expression, are showed by line charts (SPL, red line; 23CA, black line; linear regressions, dotted lines).

**Figure 7 cancers-11-00575-f007:**
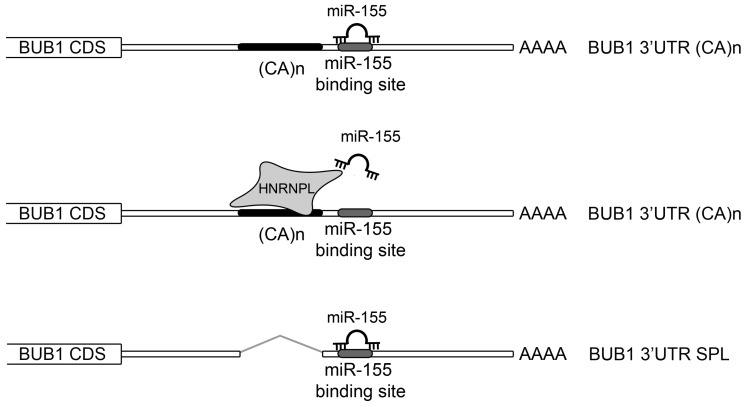
Graphical representation of the proposed model of *miR-155*/HNRNPL/BUB1 axis. The targeting of BUB1 by *miR-155* could be regulated by the binding of HNRNPL to the CA repeats in the *BUB1* 3′UTR, hindering the availability of the miR-155 binding site. In the shorter form of the 3′UTR of *BUB1*, HNRNPL cannot bound the 3′UTR, permitting the post-transcriptional regulation of BUB1 by the *miR-155*.

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
