# Peer review of "HNRNPL Restrains miR-155 Targeting of BUB1 to Stabilize Aberrant Karyotypes of Transformed Cells in Chronic Lymphocytic Leukemia"

_cancers, 2019, doi:10.3390/cancers11040575_

Reviewer 1 Report

In this manuscript by S. Pagotto et al., the authors found that higher BUB1 levels were sometimes established during immortalization of human normal dermal fibroblasts. This observation opposed the mechanism of miR-155 (elevated during immortalization) targeting BUB1 mRNA to repress BUB1 protein expression. This unexpected effect was analyzed using CLL patient samples and a CLL cell line and was found to involve hnRNP L binding to a polymorphic CA-repeat sequence in the 3’ UTR of the BUB1 transcript and inhibiting binding of miR-155 to a neighboring site in the 3’ UTR. Interestingly, and supporting this mechanism, the authors identified what they define as a splice variant of the 3’ UTR in which the CA-repeat sequence was deleted but the miR-155 target site was retained. This variant was found to be highly sensitive to miR-155-mediated repression. Overall, this is an interesting manuscript that increases knowledge of miR-155-mediated repression of BUB1 in relation to cancer. The following comments may help strengthen and clarify points in the manuscript.

The introduction should be expanding to describe the role of BUB1 at the spindle checkpoint, state that miR-155 targets BUB1, and describe the functions of hnRNP L.

The authors state the multistep immortalization of HDFLT/nTERT cell occurred between P14 and P24. How was immortalization assessed?

While the observations are interesting, it is unclear as to whether changes in abnormal karyotypes and specifically abnormal metaphases are significant, as data are from a single experiments. The authors show western blots for two other experiments, indicating that the experiment was performed other time (Fig. S1).

The authors should describe how they scored abnormal metaphases.

From the fishplot, there appear to be clones that lose chromosome 2, 21 and 19, which might affect expression of BUB1, miR-155 and hnRNP L. The authors might want to address this possibility. Could such a clone-affect account for the lack of BUB1 elevation in one the immortalization assays?

Pertaining to the BUB1 western blots, the authors probably should have included a lane with extracts from un-manipulated HDF cells to gauge normal levels of BUB1. Also the authors should address why there are sometimes one and sometimes two bands for BUB1. If this phenomenon related to splicing, it might be an effect of hnRNP L, as that is one of its functions when binding 3’ UTRs.

The authors’ interpretation of the western blots results for MSH6 and MLH1 are not entirely convincing; especially when comparing them to the BUB1 western blot in Figure S1A.

A citation is needed for D2S1888 serving as a polymorphic marker.

On line 105, the statement, “miR-155 is able to induce microsatellite instability (MSI) in colorectal cancer,” needs more information to serve as a clear justification.

On line 112, add “the” between, “…in other…”.

On line 117-118, it is unclear as to whether the two missense mutations out of 30 samples indicative BUB1 being affected in CLL or an alternative mechanism affecting BUB1 in CLL?

The missense mutations were identified from sequencing PBMC, so it is unclear whether they represent mutations in the leukemic B-cells. This is relevant because in subsequent analyses, the authors sort the different cell populations.

In Figure 4B, while the BUB1 protein level decreased the transcript level did not. Based on this effect the authors should comment on how miR-155 represses BUB1 expression.

In figure 5, the authors identify a small 3’ UTR amplicon. They should probably be consistent in stating the length of this fragment 160/180 bp. The authors describe this fragment as splice variant of the 3’ UTR. Considering the mechanism of splicing, is this fragment more likely to be due to a genomic deletion?

Author Response

Open Review 1

Comments and Suggestions for Authors

In this manuscript by S. Pagotto et al., the authors found that higher BUB1 levels were sometimes established during immortalization of human normal dermal fibroblasts. This observation opposed the mechanism of miR-155 (elevated during immortalization) targeting BUB1 mRNA to repress BUB1 protein expression. This unexpected effect was analyzed using CLL patient samples and a CLL cell line and was found to involve hnRNP L binding to a polymorphic CA-repeat sequence in the 3’ UTR of the BUB1 transcript and inhibiting binding of miR-155 to a neighboring site in the 3’ UTR. Interestingly, and supporting this mechanism, the authors identified what they define as a splice variant of the 3’ UTR in which the CA-repeat sequence was deleted but the miR-155 target site was retained. This variant was found to be highly sensitive to miR-155-mediated repression. Overall, this is an interesting manuscript that increases knowledge of miR-155-mediated repression of BUB1 in relation to cancer. The following comments may help strengthen and clarify points in the manuscript.

R1-1. The introduction should be expanding to describe the role of BUB1 at the spindle checkpoint, state that miR-155 targets BUB1, and describe the functions of hnRNP L.

Author response: As suggested by the reviewer we expanded in the introduction section the description of BUB1 in the SAC, the miR-155targeting of BUB1 and a brief description of the function of HNRNPL in relationship to microRNA targeting:

“…We demonstrated that miR-155 targets BUB1, CENPF and ZW10 [1]; proteins implicated in preventing anaphase onset in case of unattached kinetochores [2,3]. BUB1, the most studied of the three, is a serine/threonine kinase localized at the outer plate of kinetochore whose main function is to ensure the correct chromosome alignment of the metaphase plate [3]. Indeed, BUB1 deregulation is strongly associated to aneuploidy in mammalian cells [4-6]…. .…We also investigated the mechanisms that could interferes with miR-155/BUB1 targeting, such as the involvement of HNRNPL, the RNA-binding protein that hampers the miRNAs action by binding the target site on the mRNA [7,8]

R1-2. The authors state the multistep immortalization of HDFLT/nTERT cell occurred between P14 and P24. How was immortalization assessed?

Author response: We thank the reviewer for gave us the opportunity to improve the clarity of this point. We changed the sentence to make clear that we studied few passages of the immortalization process of HDF cells.

“To explore at which extent miR-155 affects aneuploidy in more advanced stages of transformation, we analyzed chromosomes at different passages during the immortalization process of normal HDF cells (HDFLT/hTERT cells at passages P16, P20 and P24”)

Regarding the methodology of the immortalization process: we transformed normal human dermal fibroblast by sequential infection with retroviruses that transduce firstly the SV40 large T antigen (puromycin resistance) and secondly the catalytic subunit of human telomerase (hTERT) (hygromycin resistance). All the HDFLT/nTERT cells used in the work were selected for 7 days with puromycin (0.5 µg/ml) and hygromycin (50 µg/ml), as described in the paper. We growth healthy cells until passage 34 when normally primary human dermal fibroblast undergoes a limited cell division, (almost 10 passages) before entering in a senescence state.

For more accuracy, we added in the material section a brief description of the immortalization steps performed as described in Pagotto et al. [1]:

Briefly, normal HDF cells were infected at passage #4 (P4), with the retrovirus transducing the SV40 Large T antigen for 3 hours, then they were infected overnight with LV pmiRZip AS miR-155 and LV pmiRZip CTRL (System Biosciences). Then, the cells were selected for 7 days with puromycin (0.5 μg/ml) (Santa Cruz Biotechnology. After selection the cells were infected at passage #6 (P6) with the retrovirus transducing the hTERT gene for 3 hours.  Then hygromycin selection was performed FOR ADDITIONAL 7 DAYS (50 μg/ml) (InvivoGen). Since LV AS CTRL contains the copGFP reporter gene, GFP-positive cells were sorted using the FACSAriaIII, 100 μm nozzle (BD Biosciences).”

R1-3. While the observations are interesting, it is unclear as to whether changes in abnormal karyotypes and specifically abnormal metaphases are significant, as data are from a single experiments. The authors show western blots for two other experiments, indicating that the experiment was performed other time (Fig. S1).

Author response: See R2.1 author response

R1-4. The authors should describe how they scored abnormal metaphases.

Author response: As requested by the reviewer we added in the main test, as well as in the figure legend (Fig.1B), the information needed.

Main test: “We next evaluated the morphology of the metaphase plates of the same cells by immunofluorescence, considering abnormal those metaphases that showed chromosomes with distinct spindle-positioning defects and incomplete congression [1]. We..

Figure 1B Legend: “….We considered abnormal those metaphases that showed errors in chromosomes congression with distinct defects of the kinetochore alignment respect to the metaphase plate.”

R1-5. From the fishplot, there appear to be clones that lose chromosome 2, 21 and 19, which might affect expression of BUB1, miR-155 and hnRNP L. The authors might want to address this possibility. Could such a clone-affect account for the lack of BUB1 elevation in one the immortalization assays?

            Author response:  The point made by the Reviewer is very pertinent, however counting the specific aberration events where they mostly occur (P16 and P20), we register an 10.5% of cells with chromosome 2 or 2q deletions at P20; 3% of cells with del21 at P20; 2% of cells with chr21 amplification at P20; 3.2% of cells with chr19 deletion at P16. These percentages of cells presenting deletions unlikely could affect the expressions of our genes of interest. Nevertheless, the expression of BUB1, which shows the most significant portion of cells with its genetic deletion is increased in our experiments despite the genetic losses.

R1-6. Pertaining to the BUB1 western blots, the authors probably should have included a lane with extracts from un-manipulated HDF cells to gauge normal levels of BUB1. Also the authors should address why there are sometimes one and sometimes two bands for BUB1. If this phenomenon related to splicing, it might be an effect of hnRNP L, as that is one of its functions when binding 3’ UTRs.

Author response: We were interested in BUB1protein expression during the different passages of HDF cellular transformation, comparing the levels very soon after the infections (SV40LT/hTERT) and in later passages. We did not consider the BUB1 protein expression of normal HDF also because the infected cells have been treated with antibiotics for selection, which make the 2 types of cells not perfectly comparable for protein expression.

Concerning the two bands of BUB1 protein showed in Fig.1D (100KDa and 120KDa); in the course of our studies, we tested three different antibodies of BUB1: BUB1 (H-300), BUB1 (B3) (Santa Cruz Biotechnology) and BUB1 MAB3610 (Millipore). We decided to use the BUB1 (B3) antibody since i) it recognizes a protein with the predicted molecular weight of BUB1 (122 KDa); ii) the band at 122 KDa is modulated by the miR-155 expression, which we experimentally found targets the 3'UTR of BUB1 (luciferase data in U2OS, western blot in HCT116, U2OS, MCF7, HG-3, and MEC01 cell lines); iii) it was already used  in published work [1,9]; iv) it was tested for immunofluorescence; v) the clone (H-300) was discontinued;

The extra bands (different from the 122 KDa form) that appear in some WBs are justified by the Santa Cruz technical support in the “Questions section” as reported below:

" Technical service·: Thank you for your question. The listed molecular weight on the datasheet refers to a publication that cited the size for BUB1 as 150 kDa (PubMedID: 1142067), however, published weights vary as low as 110 kDa for this protein (PubMedID: 10704439). The predicted weight based on sequence for BUB1 is ~122 kDa, and it undergoes post-translational modification including multiple phosphorylation sites, which changes it's observed weight on the gel. Our data falls within the observed range of 110-150 kDa. If you have any further questions please contact our Technical Services Department."

Regarding the possible effect of HNRNPL in this wonder, as the reviewer suggests, we cannot exclude a phenomenon related to splicing ruled by HNRNPL. However, HNRNPL seems to regulate mRNA splicing by binding intronic/exonic regions of the pre-mRNA to give rise to different isoforms[10]. When it binds the 3'UTRs seems to act differently, for example by inhibiting the miRNA targeting [8,11] . If HNRNPL has a role in the production of different forms of BUB1 protein it should be due to the binding of HNRNPL at different site than that we studied. Indeed, this gene binds to different regions located in BUB1 pre-mRNA (Figure S4).

For more clarity, we showed in figure 1D the quantification of two bands, but to avoid confusion we changed it, showing only the correct one. For the same reasons, we quantified only the band of 163KDa in the MSH6 immunoblot MSH6 (Supplementary figure S2).

R1-7. The authors’ interpretation of the western blots results for MSH6 and MLH1 are not entirely convincing; especially when comparing them to the BUB1 western blot in Figure S1A.

Author response: We performed the western blots analysis of MSH6 and MLH1 in the other two experiments (Supplementary figure 2B,C), as requested by reviewer 2. As discussed in the author response to R2.3 comment, the WB analysis was performed on a heterogeneous population of HDF cells, limiting to a trend the obtained results. Comparing the densitometry of the three genes, in the first experiment BUB1 starting from p19 to p24 keeps constantly values >1(value in an early passage) with small fluctuations probably due to the heterogeneity of the cells; in the second experiment, BUB1 has exactly the same trend until p24 then it starts to decrease but its levels do not go lower those in the first analyzed point.  MLH1 and MSH6 have a completely different pattern. MLH1levels, in both experiments, is constantly equal/lower compared to the first analyzed passage while MSH6 levels increase at p20 in the first experiment and at p25 and p26 in the second experiment but generally it is not possible to identify a trend that suggests and increased expression over passages

R1-8. A citation is needed for D2S1888 serving as a polymorphic marker.

Author response: As requested by the reviewer we added in the main test the information needed to reach the information about the D2S1888 in the SNP database (rs113391170 at the https://www.ncbi.nlm.nih.gov/snp). Nothwang et al. constructed the NPHP1 gene map at 2q12-q13 chromosome region in 1998 [12]

R1-9. On line 105, the statement, “miR-155 is able to induce microsatellite instability (MSI) in colorectal cancer,” needs more information to serve as a clear justification.

            Author response:  We rephrased the sentence by adding the genes involved in colorectal MSI that are targeted by miR-155 [13] and hinted the definition of CA dinucleotide repeat as microsatellite:

“….1) miR-155 is involved in microsatellite instability (MSI) in colorectal cancer by targeting hMLH1, MSH2, and MSH6 and therefore potentially affecting the CA repeat sequence [13],….”

R1-10. On line 112, add “the” between, “…in other…”.

            Author response: The issue is fixed

R1-11. On line 117-118, it is unclear as to whether the two missense mutations out of 30 samples indicative BUB1 being affected in CLL or an alternative mechanism affecting BUB1 in CLL?

            Author response:  In this section, we listed the reasons that turned us to choose chronic lymphocytic leukemia as the model of human cancer for further investigation of the miR-155/BUB1/HNRNPL axis. The germline mutations we identified evidence, as the other considerations, the importance of BUB1 in CLL. We did not suggest any molecular mechanism of these specific mutations.

R1-12. The missense mutations were identified from sequencing PBMC, so it is unclear whether they represent mutations in the leukemic B-cells. This is relevant because in subsequent analyses, the authors sort the different cell populations.

            Author response: As the reviewer noticed, the missense mutations we found were identified in PBMC; however, technically the result suggested a nucleotide variation in heterozygosity despite the heterogeneity character of the sampling. This consideration indicates the heritability of the mutations founded, indeed the same heterozygous mutations were found in the respective swabs samples from CLL patients (figure 2). These results suggest these germline mutations as predisposing CLL risk factors. Interesting to note that markers in the 2q13 chr region were already found to be predisposing CLL risk factors [14]. In a different experiment, we sorted different cell populations to investigate possible microsatellite instability in leukemic cells rather than in T normal lymphocytes.

R1-13. In Figure 4B, while the BUB1 protein level decreased the transcript level did not. Based on this effect the authors should comment on how miR-155 represses BUB1 expression.

            Author response: On the basis of our findings, the silencing HNRNPL exposes the BUB1 3'UTR to miR-155 targeting, resulting in the silencing of BUB1 protein by microRNA action without causing BUB1 mRNA degradation [15].

R1-14. In figure 5, the authors identify a small 3’ UTR amplicon. They should probably be consistent in stating the length of this fragment 160/180 bp. The authors describe this fragment as splice variant of the 3’ UTR. Considering the mechanism of splicing, is this fragment more likely to be due to a genomic deletion?

Author response: This is an interesting point of the review. We did not consider the shorter 3'UTR as a consequence of a genomic deletion because we found this shorter 3'UTR in HG-3 cells, in several experiments of HDF immortalization, and also in several sample patients of colorectal cancer and CLL (data not shown). Considering unlikely to detect so many samples with the same deletion, we investigated on the hypothesis of a post-transcriptional maturation of the mRNA to generate the shorter BUB1 3'UTR.

The length of the spliced fragment is fixed in the main test and figures 

Reviewer 2 Report

The manuscript by Pagotto and colleagues presents interesting concepts about the link between miR-155, BUB1 and HNRNPL in the light of chromosome instability especially in the context of chronic lymphocytic leukemia. However the claims of the abstract are not fully supported by the experiments performed. Indeed, several concerns preclude the publication of this work in the present form. Here is the points to be addressed:

Major points:

- In general, it is not clear how many times each experiment were performed and what is figured on the charts. Experiments should be done at least 3 times independently and bar charts should display results of the 3 experiments with statistical analysis.

- Some WB show 2 bands for BUB1 and some don’t. Could the authors explained this discrepancy?

- The authors claims that BUB1 protein decreased during HDF passages in 2 out of 3 experiments. The WB are not really convincing. The bands are increasing or decreasing depending on the passages and the experiments. It is very difficult to conclude based on these experiments. WB analyses are performed on bulk cells with different karyotype and metaphases status so it seems difficult to really correlate BUB1 expression with the expansion of normal metaphase plates on these type of samples.

Why miR-155 expression is not showed in S1B?

- Why expression of MSH6 and MLH1 was not checked on the two other experiments? As previously the WB shown do not fully support authors’ claims.

- Again, the correlation of BUB1 protein level with CA repeat instability is observed in one out of 3 experiments which render the interpretation challenging.

- The authors described germline mutation of BUB1 in CLL samples. It is not clear how many times the mutations were found in the analyzed cohort, neither which patients were analyzed. In addition, I don’t see the relevance of this study regarding the present article.

-  The authors did not find any difference in D2S1888 marker between CLL patients and healthy controls. How do they reconcile this result with their hypothesis, i.e. with the increased expression of BUB1 related to a higher binding of HNRNPL on BUB1 3’UTR and thus less targeting by miR-155?

- Authors stated that they found a positive correlation between BUB1 and HNRNPL expression. This should be shown on a plot.

- WB of figure 3  are either performed on sorted CLL cells or on total PBMC, and mixed in the same figure. This is quite confusing and may hide some differences due to variable amount of non-malignant cells.

- Silencing of HNRPL is very low (-20% according to fig 4B). It is difficult to draw definitive conclusion based on that, especially if the experiment was performed only one time. Technical replicates are not considered as independent experiments.

- Figure 5A: real-time PCR should be performed to really quantify accurately on multiples experiments.

- Figure 5E: why no control (without the sponge) are presented? It doesn’t make sense to quantify relatively to the 2µg point.

Minor points:

- Typos or inconsistency should be checked throughout the manuscript (lines 85, 93…)

- BUB1 should be introduced in the corresponding section

Author Response

Open Review 2

Comments and Suggestions for Authors

The manuscript by Pagotto and colleagues presents interesting concepts about the link between miR-155, BUB1 and HNRNPL in the light of chromosome instability especially in the context of chronic lymphocytic leukemia. However the claims of the abstract are not fully supported by the experiments performed. Indeed, several concerns preclude the publication of this work in the present form. Here is the points to be addressed:

Major points:

R2-1. In general, it is not clear how many times each experiment were performed and what is figured on the charts. Experiments should be done at least 3 times independently and bar charts should display results of the 3 experiments with statistical analysis.

Author response: Material and methods and legends have been improved to clearly outline when the experiments have been performed 2 or 3 times when cell lines were used or on a set of primary cells from CLL patients. Where not indicated, the experiments have been performed ones and for the following reasons.

Figure 1A and B:

In this experiment HDFLT/hTERT cells at passage 16 were splitted in four different aliquots to perform karyotyping (Fig1A) immunofluorescence (Fig1B), BUB1western blot and miR-155 RT-qPCR (Fig1D).The same have been done for HDFLT/hTERT cells at passage 20 and 24.Given the complexity of the experiment and the large amount of cells to be used to not allow many replications during passages, some experiments have been performed ones because supported by the literature. These four different approaches gave us the basis to formulate the hypothesis: miR-155 loses its control on BUB1over cell passages and BUB1 increase that we observed, could be important to stabilize aberrant clones over time. We believed at these results since increase of BUB1 was observed in two out of three experiments (see also answer to R2-3) and that aneuploidy metaphases increase over passages in our cellular model is supported by the literature. In 2004 Fauth C. et al demonstrated that the aneuploidy metaphases in a similar model of human fibroblast infected with SV40LT and then hTERT, as ours, expand from 18.7% in early passages (21-27) to 33.3% in mid passages (33-43p). Although the analyzed passages are not the same, the literature supported that aberrant karyotype increase over time in these models. Several studies demonstrated the needs to stabilize the aneuploidy genome of the cancer cells to permit cancer expansion: hTERT enhance genome stability [16], gene important for the immortalization of HDF cells; the saltationist theory of cancer evolution described by Markowetz that sustain a phase of genomic stability after a phase of genomic crisis. [17];and  the Karyotypic stability in CLL [18] that underlie how key genomic aberrations in CLL are maintained over time.

Figure 4A: the binding of the HNRNPL to the CA repeats located in the 3’UTR of BUB1 has already been proved in lymphoid cells (CD4 T cells), as we reported in the manuscript (Figure S4). This data is partially new, it only demonstrates that this also occurs in B cells and has been done independently with two vectors carrying either 18 or 19 CA repeats. To better highlight this point, we rephrased the sentence in line 159 adding the part underlined:

To determine if HNRNPL interacts with BUB1 3’UTR in B-CLL cells as occur in T cells (Supplemental figure S4)”.

Moreover, also the data that HNRPNL binds with higher affinity to UTR including higher CA repeats this has been proved as we stated in line 129 (version: cancers-478124-original) we only show that this happens also in B-CLL line. To make clearer this point we rephrased the sentence in line 159 adding the part underlined:

“This analysis showed that specific binding of HNRNPL to the untranslated region of BUB1 RNA was stronger for 19CA than 18CA as previously reported in a different cellular context”.

Figure 5E (new figure 6B): We obtained the same results doing the experiments one times in 2 different cell lines instead that performing the same experiment 2 times in the same cell line.

R2-2. Some WB show 2 bands for BUB1 and some don’t. Could the authors explained this discrepancy?

Author response: See R1.6

R2-3. The authors claims that BUB1 protein decreased during HDF passages in 2 out of 3 experiments. The WB are not really convincing. The bands are increasing or decreasing depending on the passages and the experiments. It is very difficult to conclude based on these experiments. WB analyses are performed on bulk cells with different karyotype and metaphases status so it seems difficult to really correlate BUB1 expression with the expansion of normal metaphase plates on these type of samples.

Author response: As the review correctly remarks, the analysis is performed on a heterogeneous population of HDF cells, some of them will die whereas others will be selected. Therefore, the resulted data cannot be simplified by an undoubted increase of BUB1 expression but in a resulting positive trend as we registered in our HDF model

See also R1.7

R2-4. Why miR-155 expression is not showed in S1B?

Author response: As requested by the reviewer we added in supplementary figure 1B the data regarding miR-155 expression.

R2-5. Why expression of MSH6 and MLH1 was not checked on the two other experiments? As previously the WB shown do not fully support authors’ claims.

Author response: As requested by the reviewer we added in supplementary figure 2B and C the expression of MSH6 and MLH1 related to the other two other experiments. For the second question see answer to R1.7

R2-6. Again, the correlation of BUB1 protein level with CA repeat instability is observed in one out of 3 experiments which render the interpretation challenging.

Author response: BUB1 increased over time during the immortalization of HDF in two out of three experiments. We speculated a possible connection between miR-155 and BUB1 in one of these: we hint that by targeting MLH1 and MSH6 during immortalization the miR-155 was able to induce MSI at the BUB1 locus impairing its own targeting of BUB1. We exclude technical or biological artifacts in this experiment since we see instability of the marker in three passages (P19, P20, P25) suggesting that this could influence the expression of BUB1. However, we do not point out at this mechanism as much relevant in our models, indeed, after investigation in CLL samples we do not find any change in D2S1888 marker among the analyzed samples. This first experiment allowed us to investigate other mechanisms, such as the miR-155/BUB1/HNRNPL signaling in the different cellular context of CLL were MSI of the D2S1888 was not found. Several mechanisms could regulate this gene in the same context, and one can take over on the others for many reasons. One example could be our second experiment in which the rescue of BUB1 expression occurred by a mechanism that remain not identified yet.

R2-7. The authors described germline mutation of BUB1 in CLL samples. It is not clear how many times the mutations were found in the analyzed cohort, neither which patients were analyzed. In addition, I don’t see the relevance of this study regarding the present article.

Author response: We thank the reviewer for giving us the possibility to clarify this point: We sequenced the BUB1 genomic DNA of 30 PBMC samples from 30 CLL patients carrying the 11q deletion, and we found two heterozygous germline missense nucleotide variations, one in the CLL patients 439 and the other in CLL patients 459. Regarding the relevance of these findings, mutations in BUB1gene are described to be relevant in several types of cancers [4,5,19-21], including lymphoproliferative disorders. Since we found BUB1 mutation also in CLL we show this result; we believed that this together with the other evidence (lines116-119) are important to start to study the involvement of BUB1 in chronic lymphocytic leukemia.

R2-8. The authors did not find any difference in D2S1888 marker between CLL patients and healthy controls. How do they reconcile this result with their hypothesis, i.e. with the increased expression of BUB1 related to a higher binding of HNRNPL on BUB1 3’UTR and thus less targeting by miR-155?

Author response: We thank the reviewer for the opportunity to improve the clarity of this point. As pointed in the answer related to the R2.6 point, we registered an increase of BUB1 expression at late passages in 2 out of 3 HDF immortalization experiments, highlighting the role of this enzyme during the latest steps of transformation of HDF cells. In the first experiment, but not in the second one, and not in CLL cells, we registered MSI of the D2S1888 marker over time hinting the players of this regulation: BUB1 3'UTR, miR-155, and HNRNPL. Then we investigate these players in CLL cells and in the HG-3 CLL cell line, showing that increasing of HNRNPL, instead of the instability of the D2S1888 marker, impairs miR-155 targeting and consequently BUB1 increments. Therefore, we found the miR-155/BUB1/HNRNPL signaling by investigating a different mechanism... For more clarity, we added at the discussion section this sentence:

Main test (discussion section): "… we supposed a possible miR-155 two-step mechanism to sustain cellular transformation: initially its up-regulation induce genetics and chromosome instability targeting genes of the mismatch repair, as hMLH1, MSH2, and MSH6 [13], and genes involved in the SAC of the cell cycle, as BUB1, CENP-F, and ZW10 [1]; then, once induced the fittest genetic abnormalities, a second step of genomic stabilization can be supported by inducing MSI at the 3'UTR of BUB1 gene to impair miR-155 targeting and restore BUB1 expression. However, this hypothesis was not verified in our CLL model that, to restore BUB1 expression, develops another mechanism or rather increases the expression of the RNA binding protein HNRNPL which binds the BUB1 3'UTR and impairs miR-155 targeting.

R2-9. Authors stated that they found a positive correlation between BUB1 and HNRNPL expression. This should be shown on a plot.

Author response: As suggested by the review we performed a correlation analysis on our CLL samples. The plotted correlation data are displayed in Supplementary figure 6A.

R2-10. WB of figure 3 are either performed on sorted CLL cells or on total PBMC, and mixed in the same figure. This is quite confusing and may hide some differences due to variable amount of non-malignant cells.

Author response: As suggest by the reviewer, to avoid confusion we have regroup the western blots analysis of B-CLL sample in the upper panel followed by the group of PBMC samples. Moreover, we specify in the main test and in the figure legend when we performed the analysis on sorted CLL or on total PBMC

Main test: “However, considering the stability of D2S1888, we evaluated HNRNPL and BUB1 protein expression in longitudinal samples of PBMCs or pure B-CLL cells from 13 CLL patients (Table S2). When content of B cells was low or different between the sequential PBMC samples (CD5/CD19 < 70% of PBMC) B cells were purified.

(Figure 3) legend: “Western blots of BUB1 and HNRNPL proteins and RT-qPCR of miR-155 in longitudinal samples of purified B-CLL cells (LL05, LLC21, LLC34, LLC47, LLC76) and PBMC with > 70% of CD5/CD19 positive cells (LLC31, LLC52, LLC62, LLC70, LLC81, LLC87, LLC98, LLC108) from CLL patients.

R2-11. Silencing of HNRPL is very low (-20% according to fig 4B). It is difficult to draw definitive conclusion based on that, especially if the experiment was performed only one time. Technical replicates are not considered as independent experiments.

Author response: As suggested by the reviewer we performed additional experiments to strengthen our data. We replicated the western blot and the RT-qPCR regarding the experiment displayed in figure 4B. We added the new data in Supplementary Figure S6B.

R2-12. Figure 5A: real-time PCR should be performed to really quantify accurately on multiples experiments.

Author response: We thank the reviewer for notice that. Indeed, our data cannot support the increase of the alternative BUB1 3'UTR after depletion of HNRNPL but only the presence of the shorter 3'UTR fragment. We were not interested at this stage of the research to study the regulation of the spliced form by HNRNPL or other factors. Therefore, the main test was modified as followed:

Main test:"...We noticed an additional shorter PCR product of about 160bp in both HNRNPL-depleted and control cells (Figure 5A). "

Then to ensure the existence of this spliced form, based on the sequence obtained from the cloned 160bp extra-band (Figure 5C), we designed a forward oligonucleotide overlapping the 5’ site of splicing. After testing its specificity, we analyzed by RT-PCR the expression of this specific form in the PBMC samples from 4 CLL patients. We found its expression in all the samples analyzed, as well as in HG-3 and HDF cells.

We added in the main test: “…To further establish the presence of this spliced variant, we designed specific PCR primers (u58_F-4090_R; Figure 5C; Supplementary table S3) able to identify only this form to test its expression also in 8 PBMC sequential samples from 4 different CLL patients. All the samples present the specific PCR product of the spliced 3'UTR (Figure 5D).

R2-13. Figure 5E: why no control (without the sponge) are presented? It doesn’t make sense to quantify relatively to the 2µg point.

            Author response: We thank the reviewer for the opportunity to clarify this point. We performed the experiment to sponge the HNRNPL or not using either the spliced form (without the binding element of HNRNPL) and or the full BUB1 3’UTR (with the sponging HNRNPL element). Therefore, one is the control of the other one. However, the spliced form was also able to sponge the miR-155 causing the increase expression of the BUB1 endogenous expression.

As the reviewer suggests we maintain the relative protein expression based on beta-actin values and not to the 2ug point.

To better clarify the experiment, we added a line chart that resumes the data results.

Minor points:

R2-14. Typos or inconsistency should be checked throughout the manuscript (lines 85, 93…)

Author response: The issues are fixed.

R2-15. BUB1 should be introduced in the corresponding section

Author response: We addressed this point.

Reviewer 3 Report

Pagotto et al describe an interesting cellular regulatory mechanism where HNRNPL can restrain miR-155 targeting of BUB1 to 2 stabilize aberrant karyotypes of transformed cells in CLL. The content of the article appears to be original for the description of the stated phenomenon, with high quality of presentation and amount of data.

There are several minor points:

1.       Figure 2 and Figure 5C resolution should be improved.

2.       Line 166. We identified a new splice variant…, excluding the XY locus…but retaining the mir155 binding site.

3.       Line 167. Sentence can be deleted. (We analysed by luciferase assay…)

4.       Line 176. Luciferase expression vectors, not expression luciferase vectors

5.       Line 208. Figure 5 legend: Describe abbreviations AS and NC.  

6.       Line 211. Figure 6 legend: Short description of the proposed model required.

7.       Lane 220. Replace sentence “We recently described as at the early stages of cellular transformation the miR-155 targets critical proteins of the mitotic checkpoint as BUB1, CENPF, and ZW10 to causes cellular aneuploidy” with “We described that mir-155 targets BUB-1, CENPF and ZW10 during early stages of cellular transformation and causes cellular aneuploidy.”

Author Response

Open Review 3

Comments and Suggestions for Authors

Pagotto et al describe an interesting cellular regulatory mechanism where HNRNPL can restrain miR-155 targeting of BUB1 to 2 stabilize aberrant karyotypes of transformed cells in CLL. The content of the article appears to be original for the description of the stated phenomenon, with high quality of presentation and amount of data.

There are several minor points:

R3-1. Figure 2 and Figure 5C resolution should be improved.

            Author response: The issues are fixed.

R3-2. Line 166. We identified a new splice variant…, excluding the XY locus…but retaining the mir155 binding site. (SARA)

Author response: The issue is fixed.

R3-3. Line 167. Sentence can be deleted. (We analysed by luciferase assay…)

Author response: The issue is fixed.

R3-4. Line 176. Luciferase expression vectors, not expression luciferase vectors

            Author response: The issues are fixed.

R3-5. Line 208. Figure 5 legend: Describe abbreviations AS and NC.  

Author response: The issues are fixed.

R3-6. Line 211. Figure 6 legend: Short description of the proposed model required.

Author response: The issues are fixed. We added in the legend of figure 6 (new figure 7) the following text: “Figure 6. Graphical representation of the proposed model of miR-155/HNRNPL/BUB1 axis. The targeting of BUB1 by miR-155 could be regulated by the binding of HNRNPL to the CA repeats in the BUB1 3’UTR hindering the availability of the miR-155 binding site. In the shorter form of the 3'UTR of BUB1, HNRNPL cannot bound the 3’UTR permitting the post-transcriptional regulation of BUB1 by the miR-155.”

R3-7. Lane 220. Replace sentence “We recently described as at the early stages of cellular transformation the miR-155 targets critical proteins of the mitotic checkpoint as BUB1, CENPF, and ZW10 to causes cellular aneuploidy” with “We described that mir-155 targets BUB-1, CENPF and ZW10 during early stages of cellular transformation and causes cellular aneuploidy.”

Author response: The issue is fixed.

Round  2

Reviewer 2 Report

The authors answered my comments and performed additional experiments when needed.

The overall quality of the manuscript is improved.

Style and minorspell check should be performed.